# Adaptive and non-adaptive gene expression responses in prostate cancer during androgen deprivation

Reetta Nätkin[1]*, Pasi Pennanen[2], Heimo Syvälä[2], Merja Bläuer[3], Juha Kesseli[1], Teuvo L. J. Tammela[4,5], Matti Nykter[1], Teemu J. Murtola[4,5]*

**1** Faculty of Medicine and Health Technology, Prostate Cancer Research Center, Tampere University and Tays Cancer Center, Tampere, Finland, **2** Faculty of Medicine and Health Technology, Tampere University and Tays Cancer Center, Tampere University Hospital, Tampere, Finland, **3** Tampere University Hospital and Faculty of Medicine and Health Technology, Tampere Pancreas Laboratory and Department of Gastroenterology and Alimentary Tract Surgery, Tampere University, Tampere, Finland, **4** Faculty of Medicine and Health Technology, Tampere University, Tampere, Finland, **5** Department of Urology, Tays Cancer Center, Tampere, Finland

* reetta.natkin@tuni.fi (RN); teemu.murtola@tuni.fi (TJM)

**Data Availability Statement:** Data are publicly available from the Gene Expression Omnibus repository (GEO record GSE178864).

## Abstract

Androgen deprivation therapy is the cornerstone treatment of advanced prostate cancer. Eventually prostate cancer cells overcome androgen deprivation therapy, giving rise to castration resistant prostate cancer (CRPC) characterized by increased androgen receptor (AR) activity. Understanding the cellular mechanisms leading to CRPC is needed for development of novel treatments. We used long-term cell cultures to model CRPC; a testosterone-dependent cell line (VCaP-T) and cell line adapted to grow in low testosterone (VCaP-CT). These were used to uncover persistent and adaptive responses to testosterone level. RNA was sequenced to study AR-regulated genes. Expression level changed due to testosterone depletion in 418 genes in VCaP-T (AR-associated genes). To evaluate significance for CRPC growth, we compared which of them were adaptive i.e., restored expression level in VCaP-CT. Adaptive genes were enriched to steroid metabolism, immune response and lipid metabolism. The Cancer Genome Atlas Prostate Adenocarcinoma data were used to assess the association with cancer aggressiveness and progression-free survival. Expressions of 47 AR-associated or association gaining genes were statistically significant markers for progression-free survival. These included genes related to immune response, adhesion and transport. Taken together, we identified and clinically validated multiple genes being linked with progression of prostate cancer and propose several novel risk genes. Possible use as biomarkers or therapeutic targets should be studied further.

## Introduction

Prostate cancer (PCa) is the second leading cause of cancer death among men in Western countries [1]. PCa is often diagnosed as localized disease, but 5–10% of patients present with advanced, lethal PCa. Metastases occur most commonly in the bone [2]. Despite advances in

**Funding:** This research was funded by Finnish Cancer Society (grant number 3122800092) and the Expert Responsibility Area of the Pirkanmaa Hospital District (grant number 9X032) through grants awarded to TJM. The Finnish Cultural Foundation also supported the study through funding awarded to RN. The funders had no role in study design, data collection and analysis, decision to publish, or preparation of the manuscript.

**Competing interests:** I have read the journal's policy and the authors of this manuscript have the following competing interests: Teemu J. Murtola has received lecture fees from Novartis, Janssen, Ferring, Sanofi and Bayer, and is a paid consultant for Novartis, Sanofi and Janssen. Teuvo L. J. Tammela is a paid consultant for Astellas, GSK, Pfizer, Orion Pharma and Amgen. The remaining authors declare no competing interests.

overall survival among men with PCa over the past decade, recent studies suggest that survival has not markedly improved among men with metastatic PCa [3].

PCa progression is androgen-dependent, thus the mainstay of therapy for metastatic PCa is systemic hormonal therapy that stops de novo testosterone production. However, the effect is transient; after a good initial response, PCa eventually recurs after 2–3 years despite continuous hormonal treatment. This developmental phase of cancer is termed castration resistant prostate cancer (CRPC), which remains incurable [4]. Mechanisms that underlie CRPC pathogenesis are not fully elucidated, but several studies during the last decade have shown that androgen receptor (AR) remains active even in CRPC. Several cellular and molecular alterations are related to AR activation in CRPC, including *AR* amplifications, *AR* mutations, aberrant AR co-regulator activities and AR splice variant expression. These changes may confer hypersensitivity to low androgen levels as well as facilitating antagonist to agonist conversion for first-generation AR antagonists. However, these AR modifications explain only a fraction of clinical resistance [5,6]. It was also shown that AR inhibition may lead to activation of alternative oncogenic signaling pathways that cooperatively promote cancer progression [7].

Here we simulate development of castration resistance by first creating a highly testosterone-dependent PCa cell line and then a subclone cell line developing the ability to grow at low testosterone concentrations. For this purpose, we use VCaP cells which represent bone metastasis, have amplified expression of wild-type androgen receptor and express the *TMPRSS2-ERG* fusion gene [8]. We use these cell lines to study RNA expression changes occurring due to androgen suppression and to evaluate which of these changes remain in androgen-independent cell line to understand the mechanisms driving transition to castration resistance.

## Methods

RPMI 1640, L-glutamine and antibiotic-antimycotic solution, were from Invitrogen (Carlsbad, CA, USA). Charcoal-stripped fetal bovine serum (DCC-FBS), Testosterone (#T1500) and anti-beta-actin antibody (AC-15) were obtained from Sigma-Aldrich (Saint Louis, Mo, USA). Antibody for androgen receptor (ab133273) was purchased from Abcam (Cambridge, UK). Anti-rabbit IgG horse radish peroxidase (HRP)-linked antibody were from Cell Signaling Technology Inc. (Danvers, MA, USA). Enzalutamide (MDV3100) was from SMS-Gruppen Selleck-chem (Rungsted, Denmark). Cellbind® 6-well plates were purchased from Corning (Corning, NY, USA). All other disposable cell culture materials were from Nalge Nunc International (Rochester, NY, USA).

### Cell culture and cell measurement of relative cell number

VCaP cell line (passage (p.) 15.) representing androgen sensitive prostate cancer cells was a gift from Dr. Tapio Visakorpi, University of Tampere, Finland. VCaP cells were cultured in RPMI 1640 supplemented with 10% DCC-FBS, 1% L-glutamine, 1% antibiotic-antimycotic solution and 10 nM testosterone (T) for seven months to establish T-dependent subclone VCaP-T. VCaP-T cells were then cultured in the presence of 0.1 nM T for 10 months to establish a subclone, VCaP-CT, that could grow at low T level. The VCaP-T and VCaP-CT cells used in this study were cultured for 20 months and represent p.84. and p.73., respectively.

For the cell growth studies, VCaP-T and VCaP-CT cells were seeded in 6-well plates 2 x 10⁵ cells per well and allowed to attach for 24 h before the treatments. The cells were treated with vehicle (dimethyl sulfoxide, DMSO) alone or with the indicated concentrations of T, bicalutamide or enzalutamide for seven days. Growth medium including the tested drugs was renewed every other day. After treatments, the cells were fixed and stained and the relative cell number

compared to vehicle-treated control cells was assessed with modified crystal violet staining method [9]. Absorbances at 590nm wavelength were measured with a Victor 1420 Multilabel Counter (Wallac, Turku, Finland).

## Immunoblotting

For Western blot studies, cells were seeded in 75 cm2 flasks and allowed to attach for 24 hours. After the indicated treatments, cells were subjected to protein extraction with M-PER® reagent (PIERCE, Rockford, IL, USA) modified with protease inhibitors (Complete Mini Protease inhibitor cocktail tablets (Roche Diagnostics GmbH, Indianapolis, IN, USA)). SDS-PAGE and Western blotting was performed as previously described [10] with the following modifications: Chemiluminescence of immunoreactive bands was detected by using ChemiDocTM XRS+ -equipment (Bio-Rad, Hercules CA, USA). Densitometric analysis of the immunoreactive bands was performed using the ImageJ software (provided by the National Institutes of Health, Bethesda, MD, USA). Protein densities were equalized with reference to Beta-actin (Sigma-Aldrich).

## Prostate specific antigen measurement

Prostate specific antigen (PSA) concentrations were analyzed from cell culture supernatants using Human PSA-total ELISA Kit (RAB0331; Sigma-Aldrich) according to the manufacturer's instructions. Absorbances at 450nm wavelength were measured with a Victor 1420 Multilabel Counter.

Cell growth analysis and PSA measurements were repeated three times as separate experiments. For each treatment, the mean ± SD is reported.

## RNA-sequencing data analysis

RNA was extracted using Direct-Zol Miniprep Plus Kit (Zymo Research #R2070) according to the manufacturer's instructions. Library preparation was done according to Illumina TruSeq© Stranded mRNA Sample Preparation Guide (part # 15031047). Sequencing was performed with Illumina HiSeq 3000. Altogether 12 samples were sequenced using 75 bp-long paired-end reads including 3 replicates from each VCaP-T, VCaP-T treated with 0.1 nM T for 48h, VCaP-CT and VCaP-CT treated with 10 nM T for 48h. On average 111 million paired-end reads were obtained per sample. Reads were aligned using STAR aligner [11] version 2.5.4b and Ensembl reference genome GRCh38. Genewise read counts were quantified using feature-Counts [12] version 1.6.2 and Gencode annotations release 28. Statistical analysis was done using R version 3.6.1.

Differentially expressed genes were determined with DESeq2 [13] version 1.24.0. Threshold for differentially expressed genes was set as p-value <0.01 after adjustment for multiple testing, log2 fold change >1 and absolute median difference of library-size normalized read counts >13 between two conditions. To determine e.g., the genes whose expression level was restored or maintained, or the AR-association was lost, we selected the genes that would not be considered as significantly differentially expressed ($p_{adj}$ >0.1) between the cell lines or treatments. Gene expression heat maps were generated with pheatmap version 1.0.12.

Annotations for the enrichment analysis were acquired with gage [14] version 2.34.0 functions go.gsets() and kegg.gsets() from Kyoto Encyclopedia of Genes and Genomes (KEGG) pathways database October 1, 2019 (92.0) release and Gene Ontology (GO) database October 7, 2019 release. Fisher's exact test was used to find out enriched (p-value < 0.05) pathways and terms among each gene group.

## Survival analysis

Survival analysis to estimate the clinical significance of the observed AR-associated genes was performed on the cohort of The Cancer Genome Atlas (TCGA) Prostate Adenocarcinoma data [15]. Gene expression quantification data (HTSeq—FPKM-UQ) were downloaded for a total of 551 samples of 495 PCa cases from GDC Data Portal. Also clinical data were downloaded. Progression was considered to be the first of either biochemical recurrence or new tumor event (distant metastasis / locoregional recurrence) after primary treatment (radiation therapy or androgen deprivation therapy). Time-metric was months from PCa diagnosis. Follow-up continued until disease progression or the last available contact date. Clinical baseline data included age, tumor TNM-stage at diagnosis and biopsy Gleason score. Also PSA was available but as this could have been measured at any point at diagnosis or after primary treatment, it was not included in the analysis.

Participants who did not have any information about follow up or progression were excluded from the analysis. Also participants with incomplete progression data (time or new tumor event type missing) were discarded. Participants who did not have a primary tumor sample or T-stage available were filtered out and for participants who had two primary tumor samples the average of expressions was applied in analysis. After this 458 participants with 84 progression events remained for the analysis. Median follow-up time since diagnosis was 792 days.

Multivariate Cox regression analysis was performed using survival version 2.44–1.1 coxph() function. Analysis was done for each gene separately. To find out possible effect of expression level on survival, expression level of the gene as FPKM-UQ of the primary tumor sample was used as exposure variable in the Cox regression analysis. All available clinical variables (age, TNM-stage and Gleason score at diagnosis, primary treatment) were initially included in the multivariable-adjusted model. After exclusion of statistically non-significant predictor variables the model included Gleason score and T-stage. For the analysis variables were categorized as follows: Gleason score to low to intermediate (5–7) and high (8–10), T-stage to localized (1–2) and locally advanced (3–4) and expression level to low and high stratified by the median. Regression model fit was assessed with cox.zph() function and the genes not fulfilling the proportional hazards assumption were filtered out. Fisher's exact test was used to find out correlations between categorized Gleason score and gene expression level as well as T-stage and gene expression level. Multiple testing correction for p-values was done using Benjamini-Hochberg [16] method.

## Results

### Effect of testosterone on relative cell number in VCaP-T and VCaP-CT cells

VCaP cells (p.15.) were cultured in long-term in a culture medium supplemented with 10% DCC-FBS and 10 nM T. After 23 passages the growth rate of these cells (VCaP-T, p.38.) was markedly enhanced, thus assumed as androgen-sensitive. The cells were then subjected to 0.1 nM T to model the physiological conditions of lowered level of T after castration since even during androgen deprivation therapy by castration T remains detectable at very low levels. After two months of ensuing very poor growth, cell growth rate gradually started to recover, eventually reaching the level of VCaP-T cells, marking creation of castration resistant (VCaP-CT) cell line (Fig 1A). Changes in relative cell number of VCaP-T and VCaP-CT cells were then assessed in the presence of low and high T concentrations. VCaP-T cells grow poorly in the presence of 0.1 nM T whereas VCaP-CT cells grow well despite low testosterone

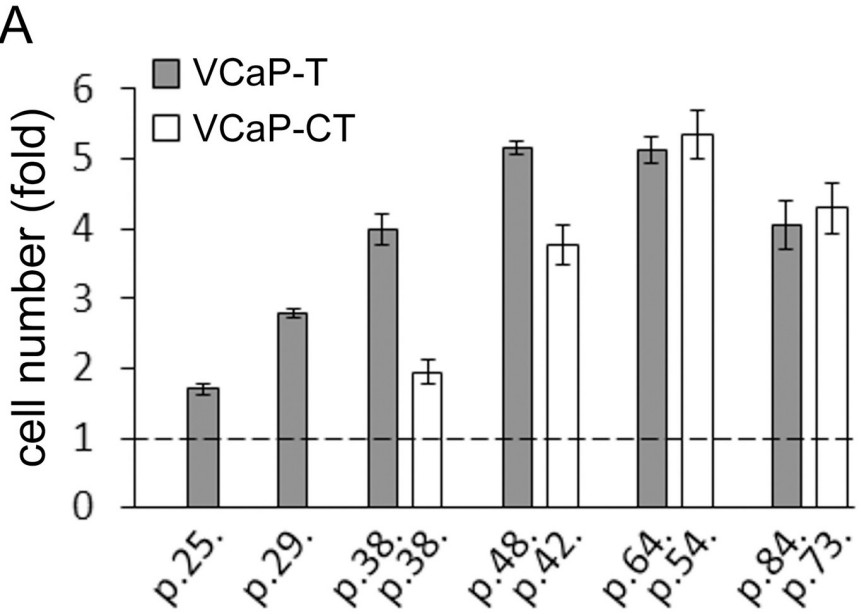

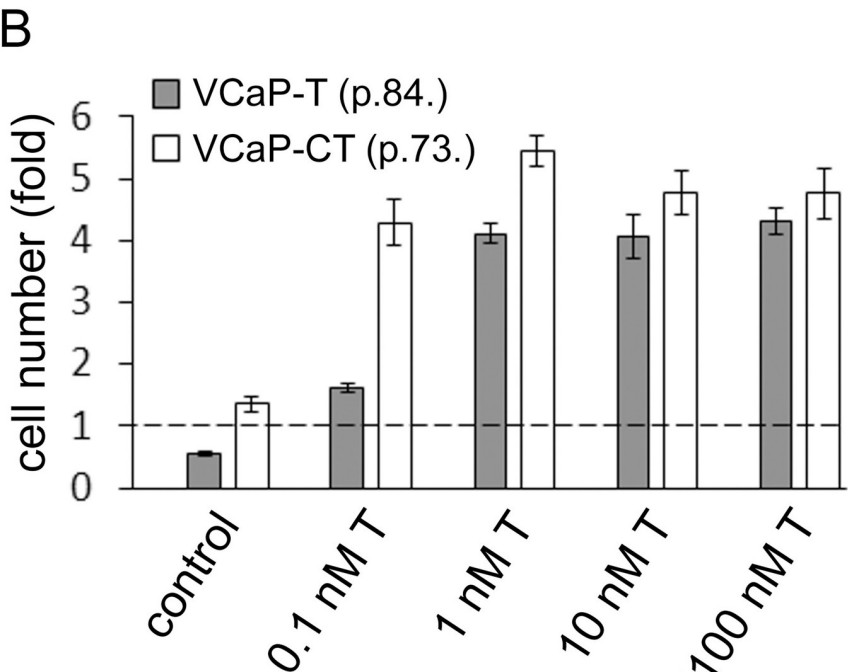

**Fig 1. Growth rates of VCaP-T and VCaP-CT during and after establishment.** (A) Growth rates of increasing passages (p.) of VCaP-T and VCaP-CT cells during long-term cell culture. (B) Growth rates of VCaP-T (p.84.) and VCaP-CT (p.73.) cells in hormone-depletion (control) and increasing nanomolar (nM) concentrations of T. The cell numbers were measured after 7 days growth and compared relative to the cell number at day 0 (dotted line). Data are represented as mean ± SD and three separate replicates were used.

concentration. However, VCaP-CT cells are still dependent on low testosterone as they cease growing in total testosterone depletion (Fig 1B).

## AR protein expression and PSA secretion

AR protein expression was detected in the presence of 0.1 nM T and 10 nM T, and additionally with antiandrogens bicalutamide or enzalutamide (Fig 2A). The expression was considered markedly changed when the density difference was more than 2-fold (S1 Fig). Full length (~100 kDa) AR protein was expressed in higher amounts in the VCaP-CT cell line than in the VCaP-T cell line. In VCaP-T cells, AR expression was not markedly changed upon different T concentrations or when the cells were additionally treated with antiandrogens. In VCaP-CT cells, AR protein expression was decreased upon testosterone addition. This down-regulation of AR was prevented when the cells were treated with T in the presence of antiandrogens.

PSA secretion was several folds higher in VCaP-T cells compared to VCaP-CT cells. It was increased by T and decreased by antiandrogens in both cell lines. However, in VCaP-CT cells the regulation by T was clearly more pronounced compared to VCaP-T (Fig 2B).

## Effect of antiandrogens on cell growth

The cell lines were treated by antiandrogens bicalutamide and enzalutamide in the presence of 0.1 nM T and 10 nM T. Antiandrogens lowered the relative cell number in both cell lines with both testosterone concentrations, although the inhibitory effect was less pronounced with the higher T concentration. Enzalutamide proved to be more potent inhibitor, especially in the presence of lower T concentration. Altogether, VCaP-CT cells were clearly more resistant to these drugs, especially to bicalutamide (Fig 3A–3D).

## RNA-sequencing

RNA sequencing (GSE178864) was used to study gene expression changes associated with exposure to low T level. First, we identified gene expression changes in VCaP-T cells after short term (48 h) testosterone depletion from 10 nM to 0.1 nM concentration to identify genes which are AR-associated. Within this set of genes altered by androgen depletion in VCaP-T, we identified genes whose expression was restored in the castration resistant VCaP-CT cells (Fig 4A). These restored genes represent candidate genes which may be AR-associated adaptive genes, and which are likely important for proliferation. Genes whose expression was not restored in VCaP-CT cells, represented AR-associated non-adaptive genes (Fig 4A). Genes whose expression did not change with short term testosterone depletion represented non-AR-associated genes (Fig 4B).

To further study AR responsiveness of genes in VCaP-CT, the cells were further treated with short term (48 h) exposure to high testosterone (10 nM). This shows which of the AR-associated genes have lost the association in castration resistance, and whether any of the non-AR-associated genes have gained AR-association in the process of adapting to low testosterone level (Fig 4A and 4B). Genes gaining the AR-association are also potentially important for CRPC progression.

## AR-associated adaptive and non-adaptive genes

We identified 418 AR-associated genes (Fig 5 and S1 Table) among which the top 5 most enriched pathways (Gene Ontology terms) were leukocyte differentiation, blood vessel development, epithelial cell morphogenesis, positive regulation of multicellular organismal process and blood vessel morphogenesis (S2 Table). Of the AR-associated genes 333 were upregulated and 85 were downregulated upon testosterone depletion.

In total 134 genes were AR-associated adaptive. Almost all (127) of them were upregulated after testosterone depletion. Enriched pathways associated with AR-associated adaptive genes

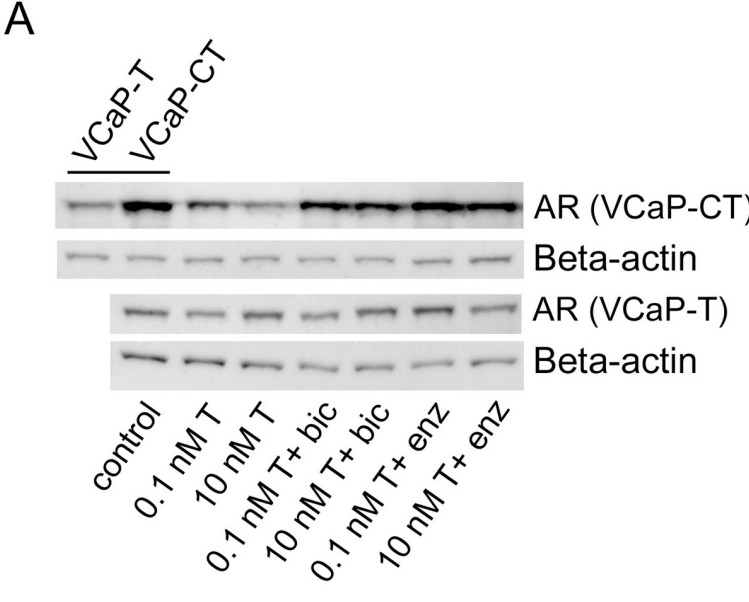

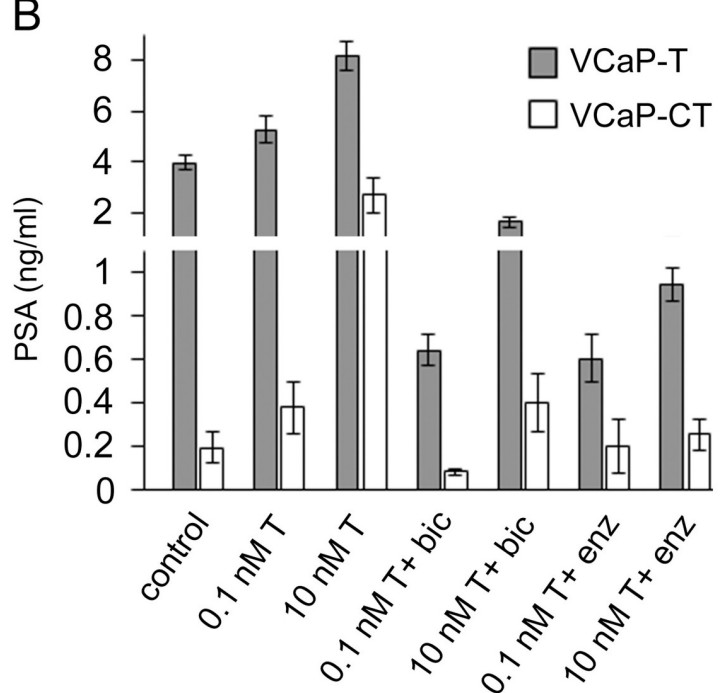

**Fig 2. Measurement of androgen receptor (AR) protein and prostate specific antigen (PSA) levels.** VCaP-T and VCaP-CT cells were treated with the indicated concentrations of testosterone (T), 10 μM bicalutamide (bic), 10 μm enzalutamide (enz) for 72 h, or were hormone-depleted for 72 h (control). (A) Analysis of AR protein levels by Western blotting. Beta-actin was used as a loading control. Control of VCaP-T (top-left corner) was additionally shown in the same Western blot as VCaP-CT to address the difference. Different rows represent different blots. Whole blots with molecular mass markers are presented in S1 Fig. (B) Analysis of the amount of secreted PSA levels. Data are represented as mean ± SD and three separate replicates were used.

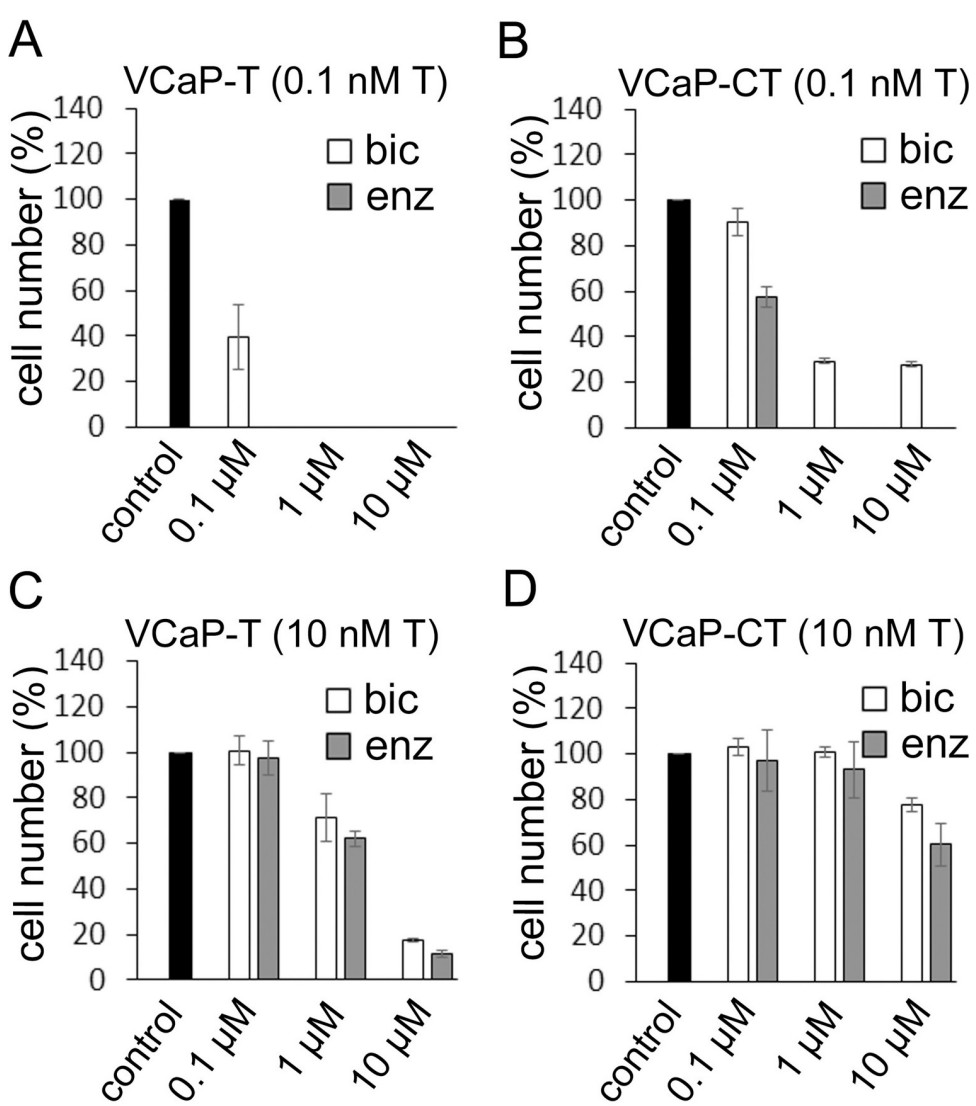

**Fig 3. The effect of antiandrogens bicalutamide or enzalutamide on the growth of VCaP-T and VCaP-CT cells.**
(A) Growth of VCaP-T in the presence of 0.1 nM T and increasing micromolar (μM) concentrations of antiandrogens. (B) Growth of VCaP-CT in the presence of 0.1 nM T and increasing concentrations of antiandrogens. (C) Growth of VCaP-T in the presence of 10 nM T and increasing concentrations of antiandrogens. (D) Growth of VCaP-CT in the presence of 10 nM T and increasing concentrations of antiandrogens. The cell numbers were measured after 7 days treatments and compared relative to the control (no antiandrogen) sample. Data are represented as mean ± SD and three separate replicates were used.

were dominated with hormone/steroid metabolism, immune system and lipid metabolism related pathways (S2 and S3 Tables). Highly significant pathways among AR-associated adaptive genes were also regulation of epithelial/endothelial cell apoptotic process.

Correspondingly 189 (127 up and 62 downregulated upon the testosterone depletion) genes were identified as AR-associated non-adaptive genes. Among AR-associated non-adaptive genes lipid metabolism, vasculature development and developmental related pathways were dominant (S2 and S3 Tables).

Regulation of lipid metabolism process was solely enriched among adaptive genes and cholesterol metabolism and fat cell differentiation were solely enriched among non-adaptive

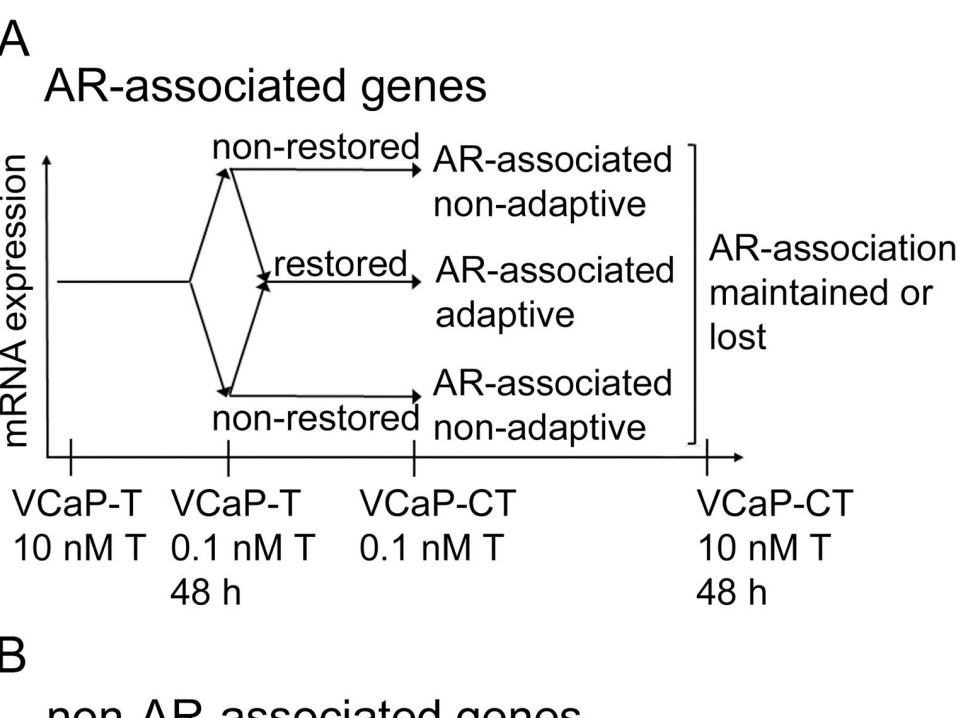

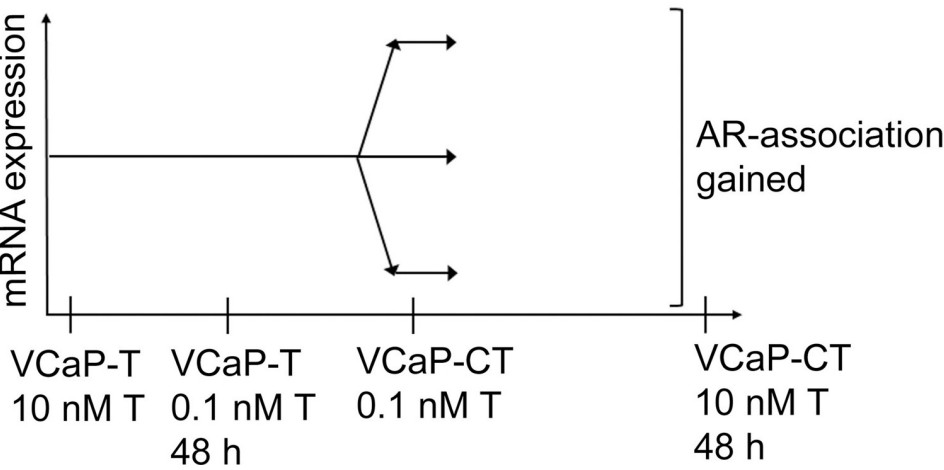

**Fig 4. Schematic illustration of the categorization of differentially expressed genes in VCaP-T and VCaP-CT cells.**
(A) AR-associated genes were identified by treating VCaP-T cells with low T concentration (0.1 nM) for 48 h. These AR-associated genes were further categorized to genes that were adaptive (expression restored) or non-adaptive (expression non-restored) in VCaP-CT cells. Whether the AR-association of the genes was lost or maintained during the acquisition to low T concentration, was assessed by treating VCaP-CT cells with 10 nM T for 48 h. (B) Genes that were not T-regulated in VCaP-T cells were categorized to non-AR-associated genes. Whether the AR-association of the genes was gained or not during the acquisition to low T concentration, was assessed by treating VCaP-CT cells with 10 nM T for 48 h.

genes. Phospholipase A2 activity and lipid transport were shared between the AR-associated adaptive and non-adaptive genes.

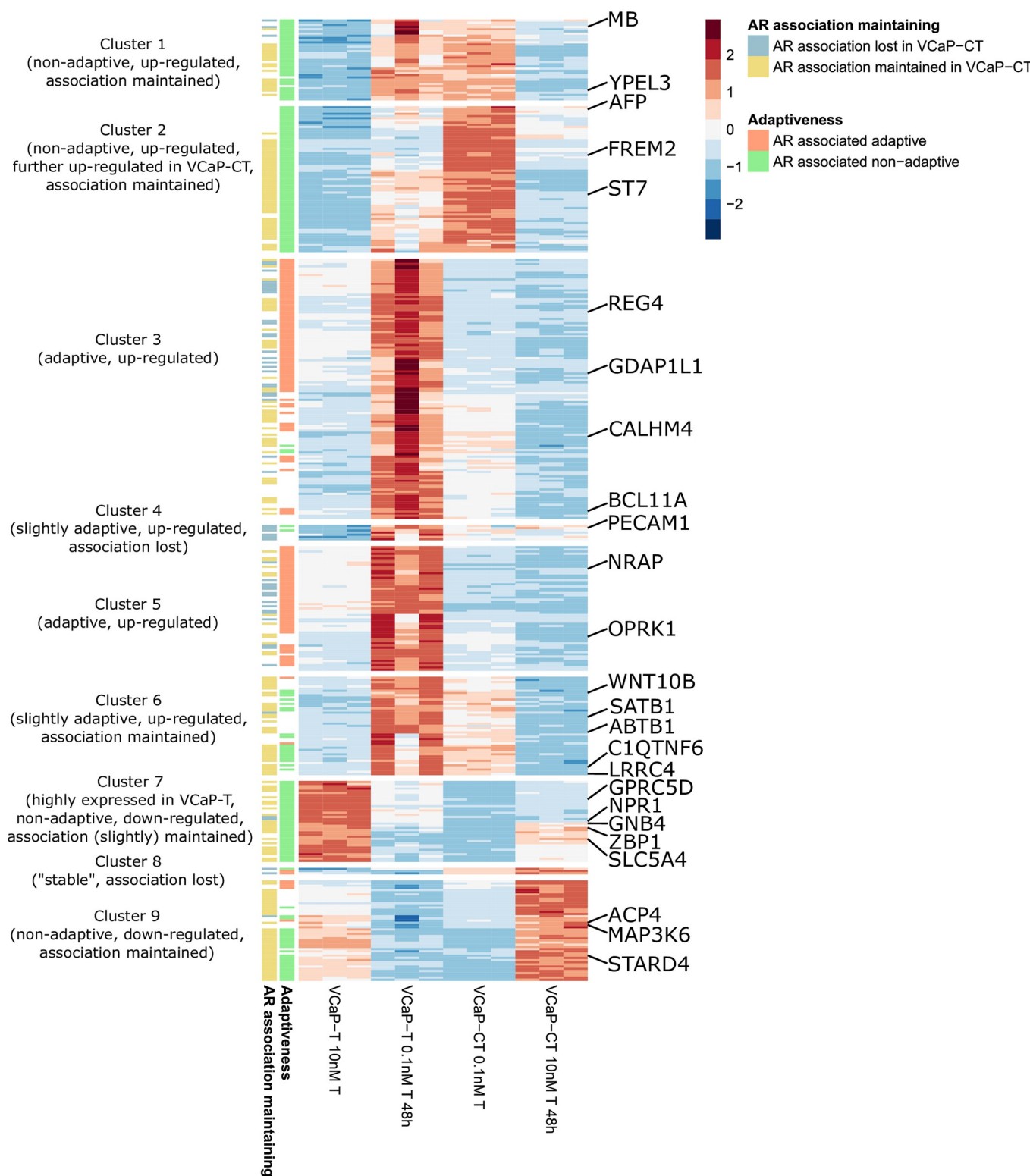

**Fig 5. AR-associated genes.** Heatmap illustration of expression levels of 418 AR-associated genes in VCaP-T and VCaP-CT cell lines upon testosterone level changes. Expression levels are scaled and centered by row. Using hierarchal clustering with Euclidean distances genes were clustered into 9 clusters with different characteristics. Genes with association to progression-free survival are pointed.

### AR-associated genes in VCaP-CT

In total 55 of AR-associated genes were identified to have lost the association in VCaP-CT cells. Comparably 214 of AR-associated genes were identified to maintain their AR-association. In total 38 adaptive and 9 non-adaptive genes lost the AR-association while 34 adaptive and 129 non-adaptive genes maintained it. NF-kappaB signaling related pathways were enriched among genes losing the AR-association (S2 and S3 Tables) and p53 signaling pathway was specifically enriched among the genes maintaining AR-association (S3 Table).

In addition to the AR-associated genes we identified 151 genes that were non-AR-associated in VCaP-T but gained the association in VCaP-CT cells (Fig 6 and S1 Table); of these 67 were upregulated and 84 downregulated after testosterone rechallenge of VCaP-CT. The most prevalent enriched pathways among this group of genes included vasculature development and cell migration/motility related pathways (S2 and S3 Tables). Also humoral immune response and inflammatory response were enriched among these genes.

### AR-associated genes as predictors of prostate cancer clinical characteristics and progression

Expression of 25 AR-associated genes and 22 genes gaining the AR-association in VCaP-CT were statistically significant predictors of risk of disease progression before correction for multiple testing (Tables 1 and S4). Notable proportion of these genes was immune system/response related, including genes *ZBP1*, *GPRC5D*, *BCL11A*, *PECAM1*, *SATB1*, *SPHK1* and *LYZ*. Also multiple genes were related to cell adhesion, including genes *PECAM1*, *LRRC4*, *FREM2*, *REG4*, *SNED1*, *NRCAM* and *SDK2*. Moreover, several genes were related to transporting, including genes *SLC5A4*, *DOC2A*, *ABCC5*, *SLC1A1*, *KCNK15*, *SYT9* and *PDGFRB*. Among lipid metabolism-related genes low expression of *STARD4* had a negative association with progression-free survival. Furthermore, majority of the genes with possible effect on PCa progression have previously been linked to cancer or cancer-related pathways. Survival associated genes were found from all clusters (Figs 5 and 6) suggesting the AR-regulation to be universal. However, none of the genes were significant predictors of disease progression after correction for multiple testing.

Gleason score showed highly significant association with the risk of disease progression. Also tumor T-stage was significantly associated with progression. We tested correlation of the expression of each gene with Gleason score and T-stage (S5 and S6 Tables). Expression of 163 genes showed significant correlation with Gleason score and expression of 124 genes showed significant correlation with T-stage. Expression of 16 genes correlating with Gleason score and 11 genes correlating with T-stage were associated also with progression-free survival (Table 1).

## Discussion

We describe development of a cell culture model to study the adaptation of androgen-dependent prostate cancer cells to low androgen level. When compared to VCaP-T cells, VCaP-CT cells were more sensitive to testosterone, gaining the ability to grow at low testosterone concentration. VCaP-CT cells reacted to testosterone-depletion by upregulating AR, which is a known mechanism of castration-resistance [17–19]. Furthermore, PSA response to testosterone was greater in testosterone-depleted VCaP-CT cells. They were also more resistant to anti-androgens, both in the presence of low and high testosterone concentration. These findings suggest both increased androgen sensitivity and responsiveness in VCaP-CT compared to VCaP-T. Thus, our findings are similar as in previously described cell culture models [19–21], confirming validity of our model in simulating castration resistance. Further, many androgen

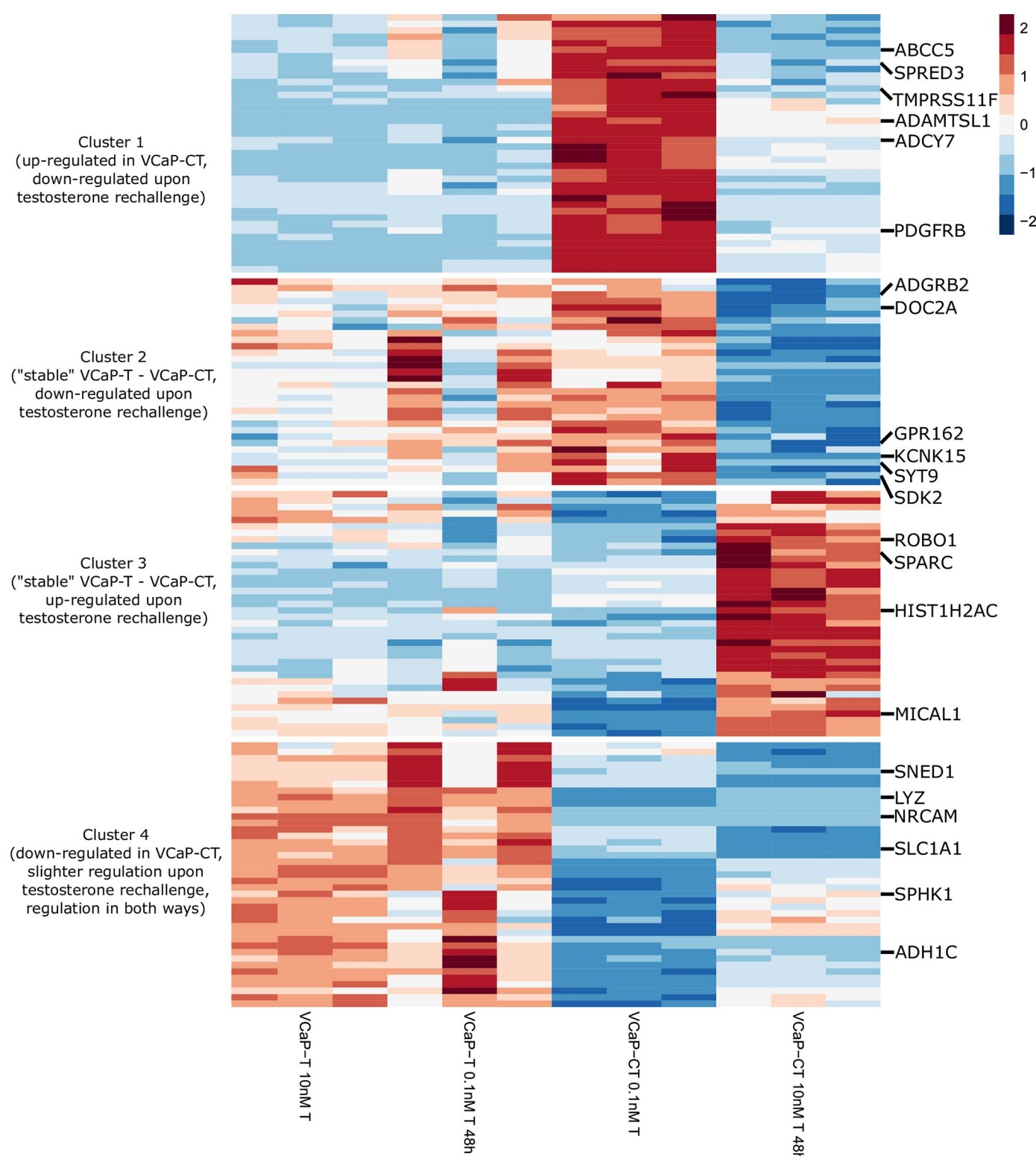

**Fig 6. Non-AR-associated genes gaining AR-association in VCaP-CT.** Heatmap of 151 non-AR-associated genes that gained AR-association in VCaP-CT and their expression levels in VCaP-T and VCaP-CT cell lines upon testosterone level changes. Expression levels are scaled and centered by row. Using hierarchal clustering with Euclidean distances genes were clustered into 4 clusters with different characteristics. Genes with association to progression-free survival are pointed.

**Table 1. Association between expression levels of androgen responsive genes and prostate cancer progression-free survival.**

| Gene | HR [1] (low vs. high expression) | 95% CI [2] | P-value | Association in VCaP-T | Adaptiveness | Association in VCaP-CT | Correlation with Gleason score[3] | Correlation with T-stage[3] |
|---|---|---|---|---|---|---|---|---|
| MICAL1 | 0.48 | 0.31–0.74 | 0.0011 | NO | | gained | - | - |
| ABTB1 | 0.46 | 0.28–0.74 | 0.0013 | YES | - | maintained | X | - |
| GPRC5D | 2.14 | 1.34–3.40 | 0.0014 | YES | non adaptive | - | X | - |
| MAP3K6 | 0.52 | 0.33–0.81 | 0.0036 | YES | - | - | - | - |
| NRAP | 1.91 | 1.22–2.99 | 0.0047 | YES | adaptive | - | - | - |
| WNT10B | 0.52 | 0.33–0.82 | 0.0050 | YES | non adaptive | maintained | - | X |
| DOC2A | 0.51 | 0.32–0.82 | 0.0052 | NO | | gained | X | - |
| BCL11A | 1.90 | 1.20–2.99 | 0.0058 | YES | adaptive | - | - | - |
| ABCC5 | 0.49 | 0.30–0.82 | 0.0061 | NO | | gained | X | X |
| ZBP1 | 0.55 | 0.35–0.85 | 0.0079 | YES | non adaptive | maintained | - | - |
| NPR1 | 0.55 | 0.35–0.86 | 0.0083 | YES | non adaptive | maintained | - | - |
| ROBO1 | 1.80 | 1.16–2.81 | 0.0092 | NO | | gained | - | - |
| GNB4 | 0.55 | 0.35–0.86 | 0.0095 | YES | non adaptive | maintained | X | - |
| SPHK1 | 0.56 | 0.36–0.87 | 0.0100 | NO | | gained | - | - |
| SPRED3 | 0.54 | 0.34–0.87 | 0.0108 | NO | | gained | X | X |
| SLC5A4 | 0.55 | 0.35–0.87 | 0.0110 | YES | non adaptive | maintained | - | - |
| ADCY7 | 0.56 | 0.36–0.88 | 0.0116 | NO | | gained | - | - |
| SNED1 | 0.58 | 0.37–0.89 | 0.0133 | NO | | gained | - | - |
| HIST1H2AC | 1.76 | 1.12–2.77 | 0.0144 | NO | | gained | - | - |
| MB | 1.73 | 1.12–2.68 | 0.0144 | YES | non adaptive | lost | - | - |
| STARD4 | 1.74 | 1.12–2.71 | 0.0146 | YES | non adaptive | maintained | - | - |
| TMPRSS11F | 1.76 | 1.11–2.80 | 0.0161 | NO | | gained | X | X |
| SPARC | 0.56 | 0.34–0.90 | 0.0173 | NO | | gained | X | X |
| PECAM1 | 0.59 | 0.37–0.92 | 0.0207 | YES | - | lost | - | - |
| ADAMTSL1 | 0.60 | 0.39–0.93 | 0.0214 | NO | | gained | - | - |
| GPR162 | 0.60 | 0.39–0.93 | 0.0237 | NO | | gained | - | - |
| LRRC4 | 0.61 | 0.39–0.94 | 0.0245 | YES | - | maintained | - | - |
| CALHM4 | 1.64 | 1.06–2.55 | 0.0273 | YES | - | - | - | - |
| ACP4 | 0.61 | 0.39–0.95 | 0.0287 | YES | - | maintained | X | - |
| ADH1C | 1.63 | 1.05–2.53 | 0.0293 | NO | | gained | - | - |
| SLC1A1 | 1.65 | 1.05–2.59 | 0.0295 | NO | | gained | X | X |
| OPRK1 | 0.62 | 0.39–0.96 | 0.0337 | YES | - | maintained | - | X |
| GDAP1L1 | 0.63 | 0.40–0.97 | 0.0357 | YES | adaptive | - | - | - |
| REG4 | 1.62 | 1.03–2.54 | 0.0383 | YES | adaptive | - | X | - |
| KCNK15 | 0.63 | 0.40–0.98 | 0.0391 | NO | | gained | - | - |
| AFP | 0.62 | 0.40–0.98 | 0.0394 | YES | non adaptive | - | - | - |
| FREM2 | 1.62 | 1.02–2.58 | 0.0398 | YES | non adaptive | maintained | X | - |
| ST7 | 1.60 | 1.02–2.52 | 0.0404 | YES | non adaptive | maintained | - | - |
| LYZ | 0.63 | 0.41–0.99 | 0.0437 | NO | | gained | - | - |
| ADGRB2 | 0.63 | 0.41–0.99 | 0.0442 | NO | | gained | - | - |
| SATB1 | 1.60 | 1.01–2.53 | 0.0449 | YES | - | maintained | X | X |
| YPEL3 | 0.64 | 0.41–0.99 | 0.0450 | YES | non adaptive | maintained | - | - |
| NRCAM | 1.58 | 1.01–2.47 | 0.0451 | NO | | gained | - | - |
| SDK2 | 0.64 | 0.42–0.99 | 0.0452 | NO | | gained | X | X |
| SYT9 | 0.64 | 0.41–0.99 | 0.0453 | NO | | gained | - | - |
| PDGFRB | 0.63 | 0.40–0.99 | 0.0474 | NO | | gained | X | X |

*(Continued)*

**Table 1.** (Continued)

| Gene | HR [1] (low vs. high expression) | 95% CI [2] | P-value | Association in VCaP-T | Adaptiveness | Association in VCaP-CT | Correlation with Gleason score[3] | Correlation with T-stage[3] |
|---|---|---|---|---|---|---|---|---|
| C1QTNF6 | 0.63 | 0.40–1.00 | 0.0488 | YES | - | maintained | X | X |

Multivariate Cox regression analysis for the genes was performed on the cohort of The Cancer Genome Atlas Prostate Adenocarcinoma data. Correlations of clinical characteristics and expression levels of the genes were assessed with Fisher's exact test.

[1] Hazard ratio for biochemical relapse or clinical tumor progression. Calculated with Cox regression.

[2] 95% confidence interval.

[3] $P \leq 0.05$ after adjustment for multiple testing.

responsive genes detected in our cell model were associated with prostate cancer progression-free survival, disease aggressiveness and tumor stage, confirming clinical validity of our findings.

RNA-sequencing revealed connections between lipid metabolism and androgen receptor regulation along with castration resistance. This agrees with previous findings about the effect of lipid metabolism to prostate cancer outcome [22–24]. Moreover, RNA expression of immune response genes was also found to be linked with AR regulation and castration resistance. Our study confirmed alteration of known prostate cancer related pathways such as steroid metabolism (e.g., *ABCA1*, *AKR1C3*, *UGT2B15*, *UGT2B17*) and epithelial/endothelial cell proliferation and apoptotic process (e.g., *CDK6*, *SNAI2*, *THBS1*, *TNFAIP3*, *EAF2*, *ATOH8*) upon AR regulation and castration resistance. Among the AR-associated genes we identified adaptive genes whose expression was restored in VCaP-CT to the level of VCaP-T under 10 nM androgen stimulus. This implicates that retaining the expression level of these genes is important for cancer cell survival and proliferation of castration resistant cells. Many of the immune response related (e.g., *IRF8*, *JAK3*, *PTGER4*, *RELB*, *IFITM3*) and part of the lipid metabolism related genes (e.g., *NR1H4*, *PPARGC1B*, *PLIN1*, *HSD17B14*) were identified to be adaptive which addressed their significance for prostate cancer.

Of individual genes with significant survival association, *Ankyrin Repeat And BTB Domain Containing 1* (*ABTB1*) has been associated with PTEN growth-suppressive signaling pathway; overexpression of *ABTB1* inhibits cell growth and progression of the cell cycle (G1/S phase) in vitro [25]. Nevertheless, *ABTB1* produces multiple protein forms and it is unknown which of the forms have the growth-inhibiting role [25]. In colorectal cancer low expression of *ABTB1* is associated with suppressed cancer cell proliferation and migration and low expression of miR-4319 targeting *ABTB1* is associated with poor prognosis [26]. Low expression of miR-4319 has been associated with poor survival also in prostate cancer [27]. We identified *ABTB1* to be up-regulated upon testosterone depletion in VCaP-T and also maintaining the AR-association in VCaP-CT. Our results suggest that low expression of *ABTB1* indicates better progression-free survival and therefore inhibition of ABTB1 may suppress cancer cell proliferation in prostate cancer.

*Microtubule associated monooxygenase, calponin and LIM domain containing 1* (*MICAL1*) has a role in cytoskeleton dynamics [28,29]. It has studied to promote breast cancer proliferation and invasion via ROS/PI3K/Akt/ERK signaling [30,31]. MICAL1 also negatively regulates apoptosis via MST-NDR signaling [32,33]. *MICAL1* has not been studied in prostate cancer but variants of *MICAL2* have progression promoting role in prostate cancer [34]. In our results *MICAL1* gained AR-association in VCaP-CT cells and its low expression associated with better progression-free survival. This suggests that MICAL1 could have similar tumor promoting roles in PCa than in other cancers.

*Special AT-Rich Sequence Binding Protein 1 / SATB homeobox 1* (*SATB1*) and *C1q and TNF related 6* (*C1QTNF6*) showed strong clinical relevance being correlated with both Gleason-score and T-stage as well as being associated with progression-free survival. Overexpression of chromatin organizing SATB1 has been widely studied to be associated with growth, metastatic potential, progression, poor survival and co-expression of multiple oncogenes in multiple cancer types including prostate cancer [35–38]. However, multiple studies associate low expression of SATB1 to poor prognosis [39–42] and apoptosis resistance [43] or do not found SATB1 to be associated with prognosis [44–46]. SATB1 is also involved in T-cell differentiation [47]. Our results determined *SATB1* to be up-regulated upon testosterone depletion in VCaP-T cells, being modestly adaptive and maintaining AR-association in VCaP-CT cells. Results support that low expression of *SATB1* is associated with poor progression-free survival and higher T-stage and Gleason-score at diagnosis. This is in contrast with previously suggested role of SATB1 as tumorigenesis promoter [35].

We identified *C1QTNF6* to be up-regulated upon testosterone depletion in VCaP-T and also maintaining its AR-association in VCaP-CT. Low expression of *C1QTNF6* associated with lower T-stage and Gleason score and better progression-free survival. *C1QTNF6* down-regulation has previously been studied to possibly increase breast cancer invasion in vitro [48] but also being overexpressed in multiple cancer types contributing to cancer proliferation, migration and possibly tumor angiogenesis [49–51]. In lung adenocarcinoma high expression of *C1QTNF6* has also been associated with poor prognosis [51]. C1QTNF6 may have anti-inflammatory role [52]. However, the role of this gene in prostate cancer is previously unknown. Our results suggest that C1QTNF6 may promote cancer proliferation and migration also in prostate cancer.

Among AR-association gained genes showing strong indications about clinical relevance, low expression of *ATP binding cassette subfamily C member 5* (*ABCC5*), *secreted protein acidic and cysteine rich* (*SPARC*) and *platelet derived growth factor receptor beta* (*PDGFRB*) is associated with better progression-free survival. ABCC5 is associated with resistance to many drugs [53–59] and it has been studied for example to promote metastasis to bone in breast cancer [60]. High expression of *ABCC5* has previously been linked with prostate cancer progression and with poor progression-free and overall survival [58]. Our results agree with previous studies about the adverse role of ABCC5. SPARC is highly associated with metastatic potential and bone metastases [61–64]. While expression of SPARC is often associated with promotion of metastases, cancer progression and poor prognosis [65–70], some studies suggest the opposite role for SPARC depending on the environment and progression/metastatic state [70–75]. SPARC may also have role in fatty acid transportation [76]. Our results support the progression promoting role of SPARC. PDGFRB and its oncogenic functions, prognostic value and inhibition have been studied widely in prostate cancer and other cancers [77–85]. Also PDGFRB has been associated with bone metastases [85–87] and in addition with angiogenesis [88,89]. Our results support the clinical significance of PDGFRB in prostate cancer progression.

Among AR-association gained genes, new candidate genes that strongly associated with aggressive tumor traits were *sprouty related EVH1 domain containing 3* (*SPRED3*), *transmembrane serine protease 11F* (*TMPRSS11F*) and *solute carrier family 1 member 1* (*SLC1A1*). SPRED3 is suggested to be a negative regulator of RAS/MAPK signaling [90–92] but has not been studied in prostate cancer. Our results do not suggest tumor suppressor role, as low expression was associated with better survival. Also *TMPRSS11F* was identified to gain AR-association in VCaP-CT cells. This particular member of the gene family has not been widely studied but it has been reported to contribute to tumor growth and poor outcome in acute myeloid leukemia patients [93]. Nevertheless, our results suggest different role for

TMPRSS11F in prostate cancer as its low expression was associated with poor progression-free survival and high Gleason score and T-stage.

Glutamate and aspartate transporting *SLC1A1* is expressed in prostate cancer and has been studied to be regulated by testosterone [94,95]. *SLC1A1* is often upregulated in cancers compared to normal tissues [96–98] but also downregulated in some cases [99] and in metastatic PCa compared to primary PCa [96]. SLC1A1 is responsible for transporting extracellular glutamate into cells and due to dysregulation, accumulation of high concentration of extracellular glutamate promotes malignant growth via glutamate receptors and activation of signaling pathways [100–102]. Contrary to that, it has been studied that SLC1A1 is needed for cystine uptake and thus prevention of oxidative stress [98,103–105]. Our results link low expression of *SLC1A1* to poor progression-free survival thus emphasizing the significance of glutamate signaling also in prostate cancer.

*Wnt family member 10B* (*WNT10B*) and *mitogen-activated protein kinase kinase kinase 6* (*MAP3K6*) associated with progression-free survival in our analysis belong to widely studied and cancer related *WNT* and *MAPK* gene families [106,107]. WNT10B promotes proliferation, invasion and other malignant functions and is associated with decreased survival in multiple cancer types [108–114]. Oncogenic functions of WNT10B have been studied also in prostate cancer [115,116] and they might be disease stage specific [116]. Our results further associates *WNT10B* with progression-free survival in PCa. Previous studies on the role of MAP3K6 in cancer are contradictory. MAP3K6 expression has been reported to be reduced in cancer and acting as a tumor suppressor by enhancing proapoptotic activities in response to stress [117–120]. Closely associated MAP3K5 has been studied to have similar role also in prostate cancer [121–123]. Still, *MAP3K6* inhibition is reported to suppress tumor growth, vessel formation and VEGF expression [124]. Our analysis associated low expression of *MAP3K6* with better progression-free survival thus suggesting other than the tumor suppressor role for MAP3K6 in PCa and that MAP3K6 inhibition could suppress tumor growth also in prostate cancer.

## Conclusion

In conclusion, in cell culture model that has gained androgen dependence in vitro and another which has gained androgen independence we confirm the role of several previously reported risk genes in castration-resistance of prostate cancer. Furthermore, we report several novel risk genes among immune-related, transport-related and cell adhesion-related genes that are also associated with disease-specific progression-free survival. We also found genes in immune response, cellular migration and motility that became androgen-regulated in castration-resistance, suggesting a role in disease progression. These findings increase knowledge on mechanisms of castration resistance and may open new avenues for treatment of this condition. Further studies are needed to understand the function and significance of reported genes as biomarkers and/or therapeutic targets in prostate cancer and castration resistance.

## Supporting information

**S1 Fig. AR protein levels by Western blotting.** Analysis of androgen receptor (AR) protein levels by Western blotting with relative density values. Whole Western blots are shown. All measured densities were corrected with corresponding Beta-actin. For both Western blots separately the density of control was set to 1 and the densities of the other lanes were calculated relative to that. More than 2-fold differences were considered to be markedly changed. (PDF)

**S1 Table. AR-associated genes.**
(XLSX)

**S2 Table. GO enrichments.**
(XLSX)

**S3 Table. KEGG enrichments.**
(XLSX)

**S4 Table. Association with progression free survival.**
(XLSX)

**S5 Table. Association with Gleason score.**
(XLSX)

**S6 Table. Association with T-stage.**
(XLSX)

**S1 Raw images. Raw images of Western blots form Fig 2A and S1 Fig.**
(PDF)

## Acknowledgments

The authors thank Niina Ikonen for excellent technical assistance. The authors wish to acknowledge the services of Finnish Functional Genomics Centre, University of Turku and Åbo Akademi and Biocenter Finland. The authors wish to acknowledge CSC–IT Center for Science, Finland, for computational resources. The results published here are in part based upon data generated by the TCGA Research Network: https://www.cancer.gov/tcga.

## Author Contributions

**Conceptualization:** Reetta Nätkin, Pasi Pennanen, Heimo Syvälä, Merja Bläuer, Teuvo L. J. Tammela, Matti Nykter, Teemu J. Murtola.

**Data curation:** Reetta Nätkin, Pasi Pennanen.

**Formal analysis:** Reetta Nätkin, Pasi Pennanen.

**Funding acquisition:** Teuvo L. J. Tammela, Teemu J. Murtola.

**Investigation:** Reetta Nätkin, Pasi Pennanen, Heimo Syvälä, Merja Bläuer, Juha Kesseli, Teuvo L. J. Tammela, Matti Nykter, Teemu J. Murtola.

**Methodology:** Reetta Nätkin, Pasi Pennanen, Heimo Syvälä, Merja Bläuer, Juha Kesseli.

**Project administration:** Teuvo L. J. Tammela, Teemu J. Murtola.

**Resources:** Teuvo L. J. Tammela, Matti Nykter, Teemu J. Murtola.

**Supervision:** Teuvo L. J. Tammela, Matti Nykter, Teemu J. Murtola.

**Validation:** Reetta Nätkin, Pasi Pennanen, Heimo Syvälä, Juha Kesseli, Matti Nykter.

**Visualization:** Reetta Nätkin, Pasi Pennanen.

**Writing – original draft:** Reetta Nätkin, Pasi Pennanen, Teemu J. Murtola.

**Writing – review & editing:** Reetta Nätkin, Pasi Pennanen, Heimo Syvälä, Merja Bläuer, Juha Kesseli, Teuvo L. J. Tammela, Matti Nykter, Teemu J. Murtola.

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
