## [Decision Letter · Decision Letter 0]

28 Sep 2022

PONE-D-22-16450Adaptive and non-adaptive gene expression responses in prostate cancer during androgen deprivationPLOS ONE

Dear Dr. Nätkin,

Thank you for submitting your manuscript to PLOS ONE. After careful consideration, we feel that it has merit but does not fully meet PLOS ONE’s publication criteria as it currently stands. Therefore, we invite you to submit a revised version of the manuscript that addresses the points raised during the review process.

ACADEMIC EDITOR: 

The characteristics of the sublines should be better described. 

We look forward to receiving your revised manuscript.

Kind regards,

Zoran Culig

Academic Editor

PLOS ONE

Journal Requirements:

"The authors thank Niina Ikonen for excellent technical assistance. This study was supported by Finnish Functional Genomics Centre, University of Turku and Åbo Akademi and Biocenter Finland. The authors wish to acknowledge CSC – IT Center for Science, Finland, for computational resources. The results published here are in part based upon data generated by the TCGA Research Network: " ext-link-type="uri" xlink:type="simple">https://www.cancer.gov/tcga"

"This research was funded by Finnish Cancer Society, grant number 3122800092 (T.J.M.) and Expert Responsibility Area of the Pirkanmaa Hospital District, grant number 9X032 (T.J.M.). The funders had no role in study design, data collection and analysis, decision to publish, or preparation of the manuscript."

3. We note that you have stated that you will provide repository information for your data at acceptance. Should your manuscript be accepted for publication, we will hold it until you provide the relevant accession numbers or DOIs necessary to access your data. If you wish to make changes to your Data Availability statement, please describe these changes in your cover letter and we will update your Data Availability statement to reflect the information you provide

In your cover letter, please note whether your blot/gel image data are in Supporting Information or posted at a public data repository, provide the repository URL if relevant, and provide specific details as to which raw blot/gel images, if any, are not available. Email us at plosone@plos.org if you have any questions

Additional Editor Comments (if provided):

The results are of interest, however there are some questions about characteristics of the cell sublines.

Reviewers' comments:

Reviewer's Responses to Questions

**Comments to the Author**

1. Is the manuscript technically sound, and do the data support the conclusions?

Reviewer #1: Partly

2. Has the statistical analysis been performed appropriately and rigorously? 

Reviewer #1: Yes

3. Have the authors made all data underlying the findings in their manuscript fully available?

Reviewer #1: Yes

4. Is the manuscript presented in an intelligible fashion and written in standard English?

Reviewer #1: Yes

5. Review Comments to the Author

Reviewer #1: The author established two sublines using VCaP cell line. One is the androgen-dependent line VCaP-T, which has been cultured under androgen-deprived conditions but supplied with 10 nM testosterone for a long time. Another one is the VCaP-CT cell line was derived from the VCaP-T cell line, which was switched to 0.1 nM testosterone for an additional 10 months. The authors then characterized the two sublines by RNA sequencing and found that several gene pathways were enriched in the VCaP-CT line, such as steroid metabolism, immune response, and lipid metabolism. They also linked AR-associated genes with the progression-free survival in the TCGA database. Overall, the authors generated novel VCaP cell lines with high and low testosterone levels. They also made some novel discoveries using these models. There are some concerns that the authors must address.

1. Please address the rationale for generating the VCaP-T and VCaP-CT cell lines. VCaP is an androgen-dependent cell line and is usually cultured in complete DMEM medium, not RPMI1640 medium. Why did the authors culture them in RPMI1640?

2. Why did the authors construct the VCaP-CT cell line in 0.1nM tetosterone but not cultured them in a complete androgen-deprived condition? Testosterone (0.1 nM) remain activate AR signaling.

3. Did the authors compare the parental VCaP cell line with these sublines. In particular, AR expression changes.

4. Testosterone need be metabolized to DHT to activate AR. What are the responses to the DHT in these sublines? Were the steroid metabolism genes, like Cyp17A1, HSD3B and AKR1C3, changed in these lines?

5. Why are the VCaP-T and VCaP-CT lines sensitive to enzalutamide treatment in Fig.3? Even 0.1 uM enzalutamide can suppress them 100 percent? What about parental VCaP cells?

6. PLOS authors have the option to publish the peer review history of their article (what does this mean?). If published, this will include your full peer review and any attached files.

Reviewer #1: **Yes: **Chengfei Liu, M.D., Ph.D.

---

## [Author Response · Author response to Decision Letter 0]

5 Dec 2022

PONE-D-22-16450

Response to Reviewers

Tampere 

6 November 2022

Dear Dr. Zoran Culig, 

Thank you for giving us the opportunity to submit a revised draft of our manuscript titled “Adaptive and non-adaptive gene expression responses in prostate cancer during androgen deprivation” to PLOS ONE. We appreciate the time and effort that you and the reviewer have dedicated to providing your valuable feedback on our manuscript. We are grateful for the insightful comments on and valuable improvements to our paper. We have been able to incorporate changes to reflect most of the suggestions provided by the reviewer. We have highlighted the changes within the marked-up manuscript. All page numbers refer to the revised manuscript file with tracked changes. 

Here is a point-by-point response first to the academic editor’s and then to the reviewer’s comments and concerns.

Comments from Academic Editor:

Comment 1: Please ensure that your manuscript meets PLOS ONE's style requirements, including those for file naming. The PLOS ONE style templates can be found at https://journals.plos.org/plosone/s/file?id=wjVg/PLOSOne_formatting_sample_main_body.pdf and https://journals.plos.org/plosone/s/file?id=ba62/PLOSOne_formatting_sample_title_authors_affiliations.pdf

Response: Thank you for the comment. After careful revision the manuscript should now meet the PLOS ONE’s style requirements described in these documents. The supporting information files are named based on these documents like “S1_Fig.pdf” and not like “S1_fig.pdf” as presented in here https://journals.plos.org/plosone/s/supporting-information. 

Comment 2: Thank you for stating the following in the Acknowledgments Section of your manuscript: "The authors thank Niina Ikonen for excellent technical assistance. This study was supported by Finnish Functional Genomics Centre, University of Turku and Åbo Akademi and Biocenter Finland. The authors wish to acknowledge CSC – IT Center for Science, Finland, for computational resources. The results published here are in part based upon data generated by the TCGA Research Network: https://www.cancer.gov/tcga" We note that you have provided funding information that is not currently declared in your Funding Statement. However, funding information should not appear in the Acknowledgments section or other areas of your manuscript. We will only publish funding information present in the Funding Statement section of the online submission form. Please remove any funding-related text from the manuscript and let us know how you would like to update your Funding Statement. Currently, your Funding Statement reads as follows: "This research was funded by Finnish Cancer Society, grant number 3122800092 (T.J.M.) and Expert Responsibility Area of the Pirkanmaa Hospital District, grant number 9X032 (T.J.M.). The funders had no role in study design, data collection and analysis, decision to publish, or preparation of the manuscript." Please include your amended statements within your cover letter; we will change the online submission form on your behalf.

Response: Thank you for this notification. However, there are no funding information in the Acknowledgements. We only wished to acknowledge the Finnish Functional Genomics Centre. Avoiding the misunderstandings, we have now updated the Acknowledgements to include “The authors wish to acknowledge the services of Finnish Functional Genomics Centre, University of Turku and Åbo Akademi and Biocenter Finland.” instead of “This study was supported by Finnish Functional Genomics Centre, University of Turku and Åbo Akademi and Biocenter Finland.” on the page 23 of the revised manuscript. Even though all funding information regarding that is present in the Funding Statement, we would like to update our Funding Statement as “This research was funded by Finnish Cancer Society, grant number 3122800092 (T.J.M.), Expert Responsibility Area of the Pirkanmaa Hospital District, grant number 9X032 (T.J.M.) and Finnish Cultural Foundation (R.N.). The funders had no role in study design, data collection and analysis, decision to publish, or preparation of the manuscript.”. 

Comment 3: We note that you have stated that you will provide repository information for your data at acceptance. Should your manuscript be accepted for publication, we will hold it until you provide the relevant accession numbers or DOIs necessary to access your data. If you wish to make changes to your Data Availability statement, please describe these changes in your cover letter and we will update your Data Availability statement to reflect the information you provide.

Response: Thank you for the comment. The Data Availability statement stands correct and does not need to be changed. The data have been stored to NCBI’s Gene Expression Omnibus and will be made fully available without restrictions at acceptance. Meanwhile, token ‘cbodygsujhyjtuz’ allows anonymous, read-only access to GEO record GSE178864 at https://www.ncbi.nlm.nih.gov/geo/query/acc.cgi?acc=GSE178864 while the record is kept private before publication.

Comment 4: PLOS ONE now requires that authors provide the original uncropped and unadjusted images underlying all blot or gel results reported in a submission’s figures or Supporting Information files. This policy and the journal’s other requirements for blot/gel reporting and figure preparation are described in detail at https://journals.plos.org/plosone/s/figures#loc-blot-and-gel-reporting-requirements and https://journals.plos.org/plosone/s/figures#loc-preparing-figures-from-image-files. When you submit your revised manuscript, please ensure that your figures adhere fully to these guidelines and provide the original underlying images for all blot or gel data reported in your submission. See the following link for instructions on providing the original image data: https://journals.plos.org/plosone/s/figures#loc-original-images-for-blots-and-gels. In your cover letter, please note whether your blot/gel image data are in Supporting Information or posted at a public data repository, provide the repository URL if relevant, and provide specific details as to which raw blot/gel images, if any, are not available. Email us at plosone@plos.org if you have any questions.

Response: We have now prepared S1 Raw Images -file including original underlying images for all blot or gel data reported in our submission. This and all the other images should adhere fully to the guidelines. S1 Raw Images -file (S1_raw_images.pdf) is uploaded as Supporting Information file and listed in the revised manuscript on the page 39. All raw blot/gel images are available. 

Comments from Reviewer:

The author established two sublines using VCaP cell line. One is the androgen-dependent line VCaP-T, which has been cultured under androgen-deprived conditions but supplied with 10 nM testosterone for a long time. Another one is the VCaP-CT cell line was derived from the VCaP-T cell line, which was switched to 0.1 nM testosterone for an additional 10 months. The authors then characterized the two sublines by RNA sequencing and found that several gene pathways were enriched in the VCaP-CT line, such as steroid metabolism, immune response, and lipid metabolism. They also linked AR-associated genes with the progression-free survival in the TCGA database. Overall, the authors generated novel VCaP cell lines with high and low testosterone levels. They also made some novel discoveries using these models. There are some concerns that the authors must address.

Comment 1: Please address the rationale for generating the VCaP-T and VCaP-CT cell lines. VCaP is an androgen-dependent cell line and is usually cultured in complete DMEM medium, not RPMI1640 medium. Why did the authors culture them in RPMI1640?

Response: Our objective was to generate by long-term culture first a highly androgen-dependent prostate cancer cell line (VCaP-T) to model clinically untreated advanced PCa (with androgen dependecy) in vitro, and then to continue to create a cell line (VCaP-CT) which was adapted to growing at a low (castrate level) testosterone concentration thus modelling transition from untreated PCa to castration resistance. 

As you stated, the VCaP cell line is usually cultivated in complete DMEM. Many laboratories have, however, used RPMI1640 for culturing VCaP cells. Also, we have been successfully using RPMI1640 for culturing these cells (See: e.g. Murtola et al., PLoS One. 2012;7(6): e39445. doi: 10.1371/journal.pone.0039445).

Comment 2: Why did the authors construct the VCaP-CT cell line in 0.1nM tetosterone but not cultured them in a complete androgen-deprived condition? Testosterone (0.1 nM) remain activate AR signaling.

Response: Thank you for pointing this out. 0.1nM testosterone is used to reflect physiological conditions during castration treatment. Clinically, testosterone remains detectable at low levels even during androgen deprivation therapy. This point has been now emphasized also in the revised manuscript on the page 8 as “… 0.1 nM T to model the physiological conditions of lowered level of T after castration since even during androgen deprivation therapy by castration T remains detectable at very low levels.” 

Comment 3: Did the authors compare the parental VCaP cell line with these sublines. In particular, AR expression changes.

Response: We did not perform any comparisons between parental VCaP cell line and the sublines. The VCaP cell line is well characterized and known to overexpress AR so we did not find it necessary to characterize the parental VCaP cell line. 

Comment 4: Testosterone need be metabolized to DHT to activate AR. What are the responses to the DHT in these sublines? Were the steroid metabolism genes, like Cyp17A1, HSD3B and AKR1C3, changed in these lines?

Response: Thank you for your suggestion. We will address effects of DHT in our future work. In our model multiple steroid metabolism genes and pathways are changed in the cell lines and in response to testosterone as stated in the manuscript on the page 18 “Our study confirmed alteration of known prostate cancer related pathways such as steroid metabolism (e.g., ABCA1, AKR1C3, UGT2B15, UGT2B17)” and as seen also from S1-S3 Tables. For example AKR1C3 is identified as AR-associated adaptive gene in our model. 

Comment 5: Why are the VCaP-T and VCaP-CT lines sensitive to enzalutamide treatment in Fig.3? Even 0.1 uM enzalutamide can suppress them 100 percent? What about parental VCaP cells?

Response: 0.1 uM enzalutamide is able to suppress only VCaP-T in 0.1nM T 100 percent since VCaP-T survives poorly at low T even without inhibition of AR signalling. The cell lines are naive to enzalutamide and VCaP-T is androgen dependent, there high sensitivity to enzalutamide is expected. VCaP cell line has previously been demonstrated to be sensitive to enzalutamide (See: e.g. Semaan et al., BMC Cancer. 2019;19, 972. doi: 10.1186/s12885-019-6185-0).

We look forward to hearing from you in due time regarding our submission and to respond to any further questions and comments you may have.

Sincerely,

Reetta Nätkin Pasi Pennanen Heimo Syvälä 

Merja Bläuer Juha Kesseli Teuvo L. J. Tammela 

Matti Nykter Teemu J. Murtola

---

## [Decision Letter · Decision Letter 1]

30 Jan 2023

Adaptive and non-adaptive gene expression responses in prostate cancer during androgen deprivation

PONE-D-22-16450R1

Dear Dr. Nätkin,

We’re pleased to inform you that your manuscript has been judged scientifically suitable for publication and will be formally accepted for publication once it meets all outstanding technical requirements.

Kind regards,

Zoran Culig

Academic Editor

PLOS ONE

Additional Editor Comments (optional):

Appropriate revisions have been made.

Reviewers' comments:

Reviewer's Responses to Questions

**Comments to the Author**

1. If the authors have adequately addressed your comments raised in a previous round of review and you feel that this manuscript is now acceptable for publication, you may indicate that here to bypass the “Comments to the Author” section, enter your conflict of interest statement in the “Confidential to Editor” section, and submit your "Accept" recommendation.

Reviewer #1: All comments have been addressed

2. Is the manuscript technically sound, and do the data support the conclusions?

Reviewer #1: Yes

3. Has the statistical analysis been performed appropriately and rigorously? 

Reviewer #1: Yes

4. Have the authors made all data underlying the findings in their manuscript fully available?

Reviewer #1: Yes

5. Is the manuscript presented in an intelligible fashion and written in standard English?

Reviewer #1: Yes

6. Review Comments to the Author

Reviewer #1: (No Response)

7. PLOS authors have the option to publish the peer review history of their article (what does this mean?). If published, this will include your full peer review and any attached files.

Reviewer #1: No

---

## [Editor Report · Acceptance letter]

10 Feb 2023

PONE-D-22-16450R1 

Adaptive and non-adaptive gene expression responses in prostate cancer during androgen deprivation 

Dear Dr. Nätkin:

I'm pleased to inform you that your manuscript has been deemed suitable for publication in PLOS ONE. Congratulations! Your manuscript is now with our production department. 

Kind regards, 

on behalf of

Dr. Zoran Culig 

Academic Editor

PLOS ONE